# Synchronous RNA conformational changes trigger ordered phase transitions in crystals

Saminathan Ramakrishnan[1,14], Jason R. Stagno [1,14], Chelsie E. Conrad[1,12], Jienyu Ding[1], Ping Yu[1], Yuba R. Bhandari[1], Yun-Tzai Lee [1], Gary Pauly[2], Oleksandr Yefanov[3], Max O. Wiedorn [3], Juraj Knoska [3,4], Dominik Oberthür[3], Thomas A. White [3], Anton Barty[3], Valerio Mariani[3], Chufeng Li[3,5], Wolfgang Brehm[3], William F. Heinz [6], Valentin Magidson[6], Stephen Lockett[6], Mark S. Hunter [7], Sébastien Boutet [7], Nadia A. Zatsepin [5,13], Xiaobing Zuo [8], Thomas D. Grant [9], Suraj Pandey[10], Marius Schmidt [10], John C. H. Spence[5], Henry N. Chapman [3,4,11] & Yun-Xing Wang [1✉]

Time-resolved studies of biomacromolecular crystals have been limited to systems involving only minute conformational changes within the same lattice. Ligand-induced changes greater than several angstroms, however, are likely to result in solid-solid phase transitions, which require a detailed understanding of the mechanistic interplay between conformational and lattice transitions. Here we report the synchronous behavior of the adenine riboswitch aptamer RNA in crystal during ligand-triggered isothermal phase transitions. Direct visualization using polarized video microscopy and atomic force microscopy shows that the RNA molecules undergo cooperative rearrangements that maintain lattice order, whose cell parameters change distinctly as a function of time. The bulk lattice order throughout the transition is further supported by time-resolved diffraction data from crystals using an X-ray free electron laser. The synchronous molecular rearrangements in crystal provide the physical basis for studying large conformational changes using time-resolved crystallography and micro/nanocrystals.

[1] Structural Biophysics Laboratory, National Cancer Institute, Frederick, MD, USA. [2] Chemical Biology Laboratory, Center for Cancer Research, National Cancer Institute, Frederick, MD, USA. [3] Center for Free-Electron Laser Science, Deutsches Elektronen-Synchrotron DESY, Hamburg, Germany. [4] Department of Physics, Universität Hamburg, Hamburg, Germany. [5] Department of Physics, Arizona State University, Tempe, AZ, USA. [6] Optical Microscopy and Analysis Laboratory, Cancer Research Technology Program, Frederick National Laboratory for Cancer Research, Frederick, MD, USA. [7] Linac Coherent Light Source, SLAC National Accelerator Laboratory, Menlo Park, CA, USA. [8] X-ray Science Division, Argonne National Laboratory, Lemont, IL, USA. [9] Department of Structural Biology, Jacobs School of Medicine and Biomedical Sciences, SUNY University at Buffalo, Buffalo, NY, USA. [10] Kenwood Interdisciplinary Research Complex Physics Department, University of Wisconsin-Milwaukee, Milwaukee, WI, USA. [11] Centre for Ultrafast Imaging, Universität Hamburg, Hamburg, Germany. [12] Present address: Huntsman Cancer Institute, University of Utah, Salt Lake City, UT, USA. [13] Present address: Department of Chemistry and Physics, ARC Centre of Excellence in Advanced Molecular Imaging, La Trobe Institute for Molecular Science, La Trobe University, Melbourne 3086 Victoria, Australia. [14] These authors contributed equally: Saminathan Ramakrishnan, Jason R. Stagno. ✉email: wangyunx@mail.nih.gov

Diffusionless solid–solid (SS) phase transitions, which are observed in both inorganic and organic crystals, are usually triggered by temperature, humidity, pressure, or mechanical stress that cause rearrangements of atoms or small molecules[1,2]. Well-known examples are shape changes in memory alloys caused by temperature variation[3], or in small molecular crystals by light exposure[4,5]. SS phase transitions have also been observed in living systems with functional roles, for example, in strain-induced tail-sheath contraction in T-even bacteriophages[6,7] and in flagella-assisted bacterial locomotion[8,9]. Mechanisms of transition are not well understood at the atomic or molecular levels due to technical challenges[10], even though they are fundamentally important to material and biological sciences.

The rapid development of the ultra-brilliant femtosecond X-ray-free electron laser (XFEL) has made it possible to directly visualize atomic motions, ligand binding and reaction intermediates, and covalent bond breakage and formation at room temperature[11–18]. All those studies involved photoactive or other proteins that undergo relatively small and local structural changes within the same crystal lattices. However, the majority of biological processes involve interactions that result in large conformational changes greater than several angstroms, which are likely to alter lattice interfaces and trigger phase transitions in crystal[14,19–22]. Whether lattice order is maintained throughout SS transitions is paramount to performing time-resolved crystallography (TRX) experiments on such systems.

Here we report the interplay of synchronous molecular conformational changes and ordered SS transitions in crystals of adenine riboswitch aptamer RNA (riboA), elucidated using a multidisciplinary approach. Due to lattice restraints, the conformational changes that occur in the crystal, which drive the crystal phase transition, take place on a timescale much greater than that which occur in solution or under physiological conditions. Direct visualization by polarized video microscopy (PVM) reveals both optical and physical manifestations of the SS transition in the riboA crystals in real time. Quantitative analysis of the birefringence intensity of the crystals throughout the transition allows for the elucidation of transition times and kinetics, and the level of molecular synchrony at the pixel level. Further direct visualization and image analysis at the molecular level using solution atomic force microscopy (AFM) reveal the distinct molecular arrangements and lattice constants at various stages of the transition. These microscopy data corroborate the TRX data that detect the difference electron density (DED) and evolution of intermediate structures. Together, these results describe in detail the SS transition in the riboA crystals, and demonstrate the molecular cooperativity associated with the large conformational changes triggered by ligand binding.

## Results and discussion

**Visualizing the SS transitions by PVM**. We observed the SS transitions in small plate-like riboA crystals (~$25 \times 25 \times 1$ μm$^3$) directly by measuring the changes in the intensity of crystal birefringence using PVM (Fig. 1a, Supplementary Fig. 1, and Supplementary Movie 1). The crystal "fractures" upon transition, suggesting a volume change. Detailed analyses were performed at the pixel level for a region of interest (ROI, Fig. 1a, yellow square) of the digitized video. The ROI, which contains ~400 million RNA molecules, is about the typical size of crystals used in the XFEL experiments[14]. The direct visualization at the pixel level shows that the crystal remains birefringent, indicating that the crystal never enters a liquid phase throughout the transition process. The time-traced transmitted light intensity vs. time of a single pixel exhibits multiphasic sigmoidal transitions with ~50% total loss in transmitted light intensity $I_i[(xy)_i, t]$ of pixel $I$ ($i = 1, 2, ..., 32,400$) (Fig. 1b, left), over the course of the entire transition. The first

derivatives $-\partial I_i[(xy)_i t)]/\partial t$ give rise to the half-width and transition time (Fig. 1b, right). Three main transitions (T1, T2, and T3) are apparent, with T2 split into two sub-transitions (T2a and T2b) in terms of transition width (Fig. 1c and Supplementary Fig. 2). T1 is sharp and centered at ~227 s, with >90% of all pixels completing transition within 5 s (transition width). The dependence of the transition time on the area sampling size is informative regarding the degree of spatiotemporal synchrony of a transition (Fig. 1d). The mean transition times of all three transitions appear relatively independent of the size of the sampled areas, with a standard deviation of ~2.5 s for T1 in particular (Fig. 1d), indicating that the transition synchrony spans the entire sampled area (Supplementary Movie 2). The PVM result shows that the transition occurs within the ordered lattices.

**Molecular order visualized using AFM**. At nanometer-scale resolution, AFM can discern the ordered arrangements of individual biomacromolecules on crystal surfaces and provide the unit-cell constants. Each AFM image in Fig. 2a depicts roughly the same $50 \times 50$ nm$^2$ area of an $ac$-face riboA crystal (Supplementary Figs. 3 and 4, see also "Methods"), equivalent to an ~$2 \times 2$ pixels square area in the PVM experiment. Unit-cell measurements identified three primary lattices, defined as apo unit cell (AUC), transition unit cell 1 (TUC1), and bound unit cell (BUC). We observed continuous, cooperative, and subtle variations in molecular surface topography (topo) for the time periods before the AUC to TUC1, and before the TUC1 to BUC rapid transitions. Those topographical variations are likely due to continuous molecular rotation and translation (Fig. 2a and Supplementary Table 1). The first transition, AUC to TUC1, is measurable by the change in the $c$-axis from 9.3 to 7.9 nm (Fig. 2b). The phase transition from TUC1 to BUC, however, is slow enough to observe an intermediate topography (Fig. 2a (image #119), b (image #119 and #120) and Supplementary Table 1). One portion of image #119 (red box) corresponds to TUC1 topo 2, which is similar to the preceding AFM image; another portion (white box) corresponds to BUC topo 1, which is similar to the succeeding AFM image. A third portion (green box), which is sandwiched in between, exhibits a topography that is distinct from either (Fig. 2a), and whose auto-correlation plot gives a $c$-axis measurement of ~12.5 nm. Mixed lattices in transition (AUC and TUC1), as well as a unique lattice (TUC2) with a $c$-axis of 12.4 nm, were observed in X-ray diffraction experiments (see the next section). Overall, the $c$-axis serves as a sensitive indicator for the multiphasic transition and is plotted vs. serial image number in real-time (Fig. 2b). The plot qualitatively parallels the PVM intensity plots (Fig. 1b). Based on the sequential appearance in both PVM and AFM, T1 observed by PVM corresponds to the AUC–TUC1 lattice transition, whereas T2 and T3 (Fig. 1c and see also Supplementary Fig. 2) are likely related to the transitions from TUC1 to TUC2, and from (TUC1/TUC2) to BUC (Fig. 2b). The AFM images and the time trace of the $c$-axis show that the crystal remains ordered throughout the transitions.

**Bulk-lattice order during the transitions**. The level of cooperativity (lattice order) throughout the SS phase transition was measurable by the quality and maximum scattering angle (resolution) of Bragg diffraction and by diffusive scattering that indicates deviation from perfect order[23–25]. Diffraction data were recorded at mixing time delays ranging from 0 to 175 s (Fig. 3 and see also Supplementary Fig. 5) and provide snapshots of bulk-lattice order over a large number of crystals, as opposed to the local imaging area in a single crystal shown by PVM and AFM during the transitions. With increasing delay time, the indexing rate for AUC falls off steeply with a coincident emergence of the pseudo-monoclinic transitional unit cell, TUC1 (Fig. 3a and Supplementary Table 2). Importantly, Bragg diffraction is maintained throughout the entire transition

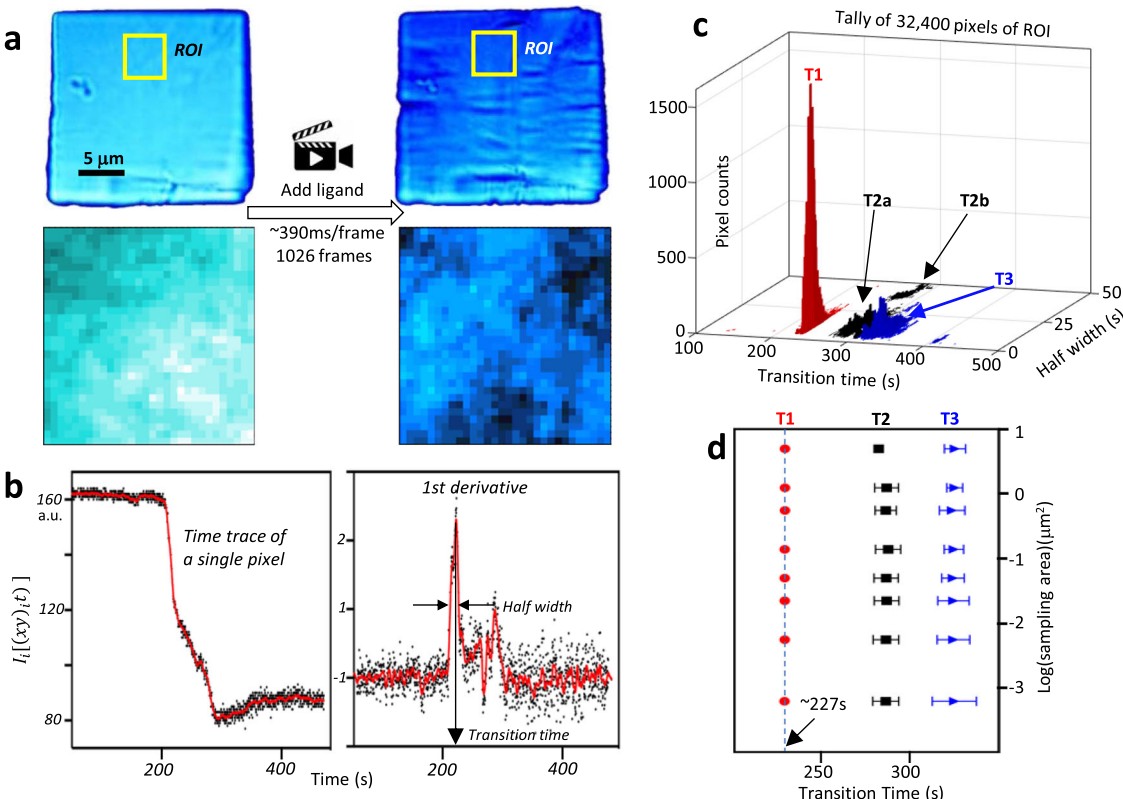

**Fig. 1 Ligand-triggered crystal lattice transition observed by PVM. a** Video microscopy images of an "ab" (c-axis along the thin edge) riboA crystal before (top left) and after (top right) the phase transition in the presence of 1 mM ligand. The yellow box defines the ~4.5 μm square (180 × 180 = 32,400 pixels) region of interest (ROI) used for the detailed analysis. The crystal is continuously monitored at the pixel level, starting from the beginning (bottom left) to the end (bottom right). A 30 × 30 pixels square area in the yellow box is used to show the crystal is birefringent at the pixel level before (bottom left) and after (bottom right) the transition. The images are color-enhanced for clear visualization of individual pixels and the nonuniform crystal surface. **b** Intensity $I_i[(xy)_i t]$ (arbitrary units) vs. time trace (in s) for the single-center pixel of the ROI. Black dots represent the intensities of individual pixels, and the red line is a time average of the derivative with a smoothing window of 15.2 s (left). Derivative $-\partial I_i[(xy)_i t)]/\partial t$ vs. time so that peaks represent the maximum rate of intensity decrease (right). Black dots represent the instantaneous derivative at each time point. The red line is a time average of the derivative with a smoothing window of 7.78 s. **c** Bivariate histogram of peak half-width of transitions 1, 2, and 3 (T1, T2a and b, and T3) (y-axis) and transition time (x-axis) for all (32,400) pixels in the ROI. **d** Transition time vs. size of sampling areas. Time zero in the PVM experiment refers to the first frame (200) used in the digitization. Source data are provided as a Source Data file.

from AUC to TUC1, with very little change in the diffraction hit-rate (Fig. 3a), and no evidence of crystals becoming completely disordered or entering the liquid phase. AUC-indexed diffraction data for each time delay (observable up to 75 s) were assembled and aligned in three-dimensional space ("3D merged") according to orientation information provided by indexing (Fig. 3b)[23,26]. In the $a^*b^*$ plane, only Bragg peaks corresponding to AUC dimensions of a (48.3 Å) and b (46.9 Å) are observed at 0 s. At 10 s, an additional set of Bragg peaks emerges corresponding to a (50.3 Å) and b (25.2 Å) dimensions of TUC1, which become more prominent with increasing delay time. The presence of peaks corresponding to TUC1 in the 3D reciprocal lattices merged with respect to AUC-indexed orientations indicates that the TUC1 diffraction comes from AUC-indexed and oriented crystals (Fig. 3b). Upon manual inspection of individual diffraction patterns, numerous images exhibited such differentiated AUC and TUC1 lattices in the same crystals (Fig. 3c), illustrating the sudden yet cooperative behavior of T1, corroborating the PVM (Fig. 1) and AFM (Fig. 2) results. In the $a^*c^*$ plane, the rapid sampling by TRX reveals azimuthal streaks about $b^*$ at 10 s, whose degree of elongation increases at 25 s (Fig. 3b), and coincides with the shortening of the c-axis (from 93.4 to 78.7 Å) and the change in the unique angle of the monoclinic cell (from β to α) (Fig. 3c). The fact that radially elongated peaks are not observed in $a^*b^*$ suggests that these features are not the result of

crystal mosaicity in the conventional sense. Instead, the molecular disorder occurs anisotropically—order is retained to a greater extent along certain directions—and is determined intrinsically by the trajectory of molecular motion and the effects it has on crystal packing. Such lattice restructuring includes both the disruption of some packing interactions and the formation of new ones, which are not necessarily isotropic or uniform in all directions.

The lattice parameters derived from the diffraction data correlate well with those measured from time-lapsed AFM data (Fig. 2b and Supplementary Table 1). In particular, the c-axis measurement (~12.5 nm) observed in the TUC1–BUC intermediate (Fig. 2a, image #119) was used to guide the indexing of diffraction data. TUC2, with a c-axis of 12.4 nm, was identified in a small population (500 crystals) of the mixing data (up to 175 s), and in a greater population (3770 crystals) in a separate pool of data collected from sample premixed with ligand (>7 min post mixing) (Supplementary Tables 3 and 4).

**Implications of synchronous phase transitions.** The ordered lattices during the transition make it possible to record large molecular conformational changes in real time. Structural changes associated with the AUC to TUC1 transition were observed in DED maps generated from AUC ligand-free (AUC-free), AUC 10

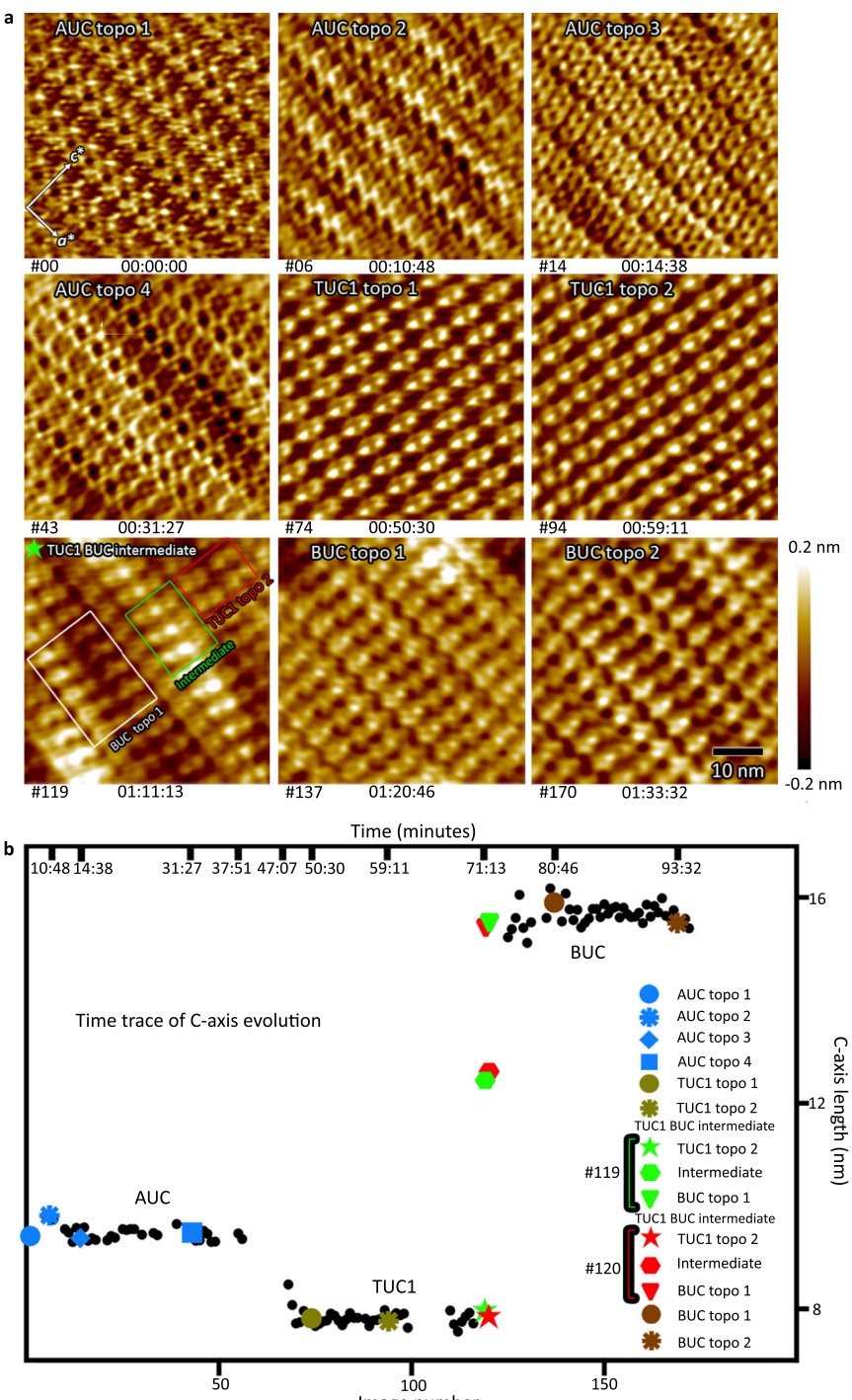

**Fig. 2 Ligand-triggered crystal lattice transition observed by AFM. a** The time-lapsed representative snapshots of the ligand-induced phase transitions in *ac* face of the riboA crystal. AUC apo unit cell, TUC1 transition unit cell 1, TUC2 transition unit cell 2, BUC bound unit cell. The order of morphological changes and phase transitions is as follows: AUC topography (topo) 1, AUC topo 2, AUC topo 3, AUC topo 4, TUC1 topo 1, TUC1 topo 2, TUC1–BUC intermediate, BUC topo 1, and BUC topo 2. **b** The plot of *c*-axis length in nanometers vs. serial image number and time (top axis) during the lattice transition. This plot summarizes the ligand-triggered polymorphic transition. Gaps in the plot are the periods when the changes in the crystal phase resulted in the loss of contact with the crystal surface during AFM imaging. Source data are provided as a Source Data file.

s-mixed (AUC-10 s), and AUC 25 s-mixed (AUC-25 s) data sets (Fig. 4 and Supplementary Table 5). For both molecules in the asymmetric unit (apo1 and apo2), large changes relative to the starting apo conformations are observed at 10 s (Fig. 4a, b, left panels). Changes in apo1 are primarily localized to the top of the 5′ strand of P1, beginning at G18, and extending through the flexible hinge region that includes A21, U22, and A23 (Fig. 4a, left). These

changes likely reflect the conversion of apo1 (ligand incompetent) toward apo2 (ligand competent), which involves large motions in the hinge and P1, including an ~180° rotation of A23 out of the binding pocket. Between 10 and 25 s, fewer changes are observed in apo1, although prominent peaks appear near latch residues 47–49, as well as G44 (Fig. 4a, right), and could be further indicative of changes toward the apo2 conformation. At 10 s, the

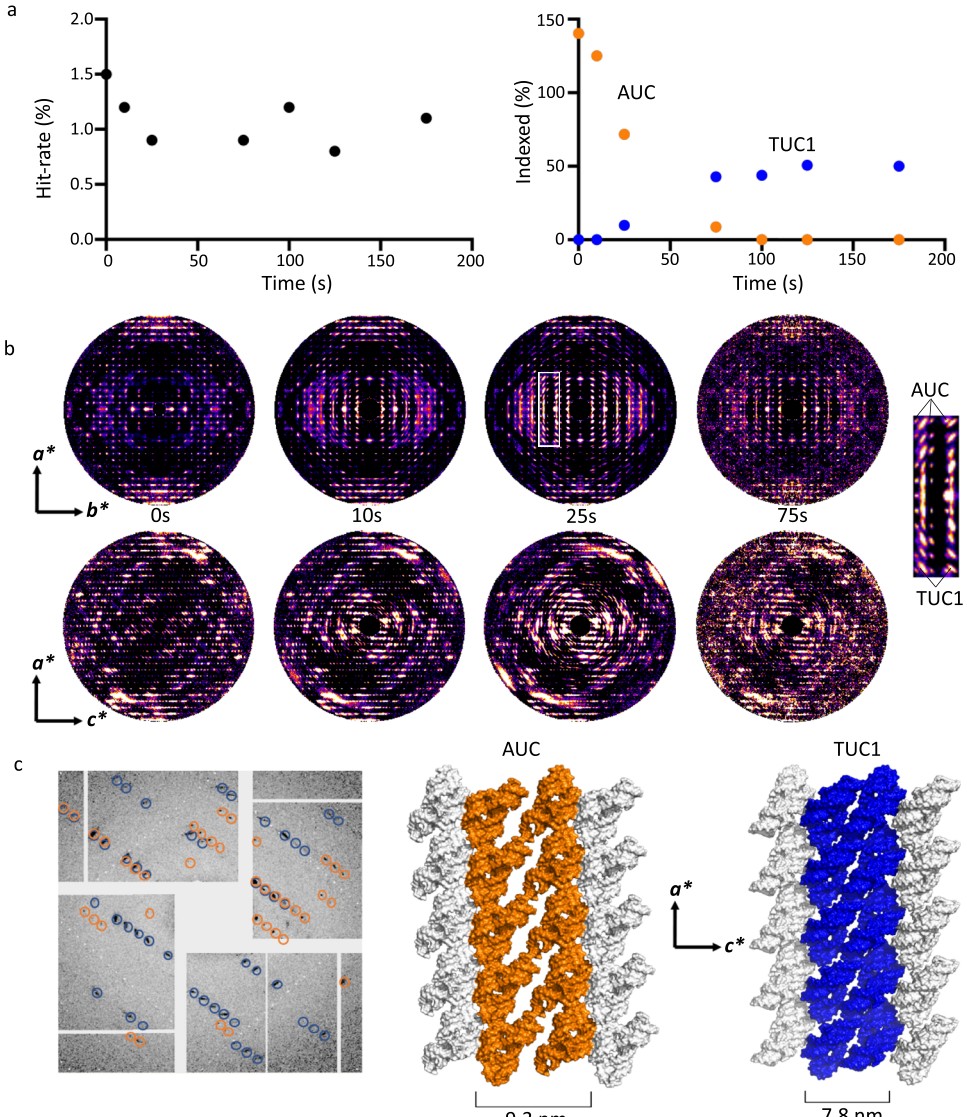

**Fig. 3 Phase transition observed by time-resolved serial crystallography. a** Plots of X-ray diffraction hit-rate (left) and indexing rate (right) for AUC (orange) and TUC1 (blue) lattices as a function of delay time after mixing 1:1 with 20 mM ADE. For the non-mixed sample (0 s), the average ratio of sample:buffer was 2:1, yielding a slightly higher hit-rate. The AUC to TUC1 conversion occurs much earlier than that observed in PVM and AFM experiments, where lower ligand concentrations (1 mM for PVM, and 50 or 150 μM for AFM) were used due to the requirements of those experiments. **b** Cross-sections ($a*b*$, $a*c*$) of the 3D-merged reciprocal lattices of AUC-indexed crystals at different delay times. An expanded subsection (white box) of $a*b*$ at 25 s shows the emergence of broad, intense Bragg peaks corresponding to the $b$-axis (25.3 Å) of TUC1. **c** Left: single diffraction image from a single crystal with circled indexed Bragg reflections belonging to either AUC (orange) or TUC1 (blue). Changes in crystal packing upon conversion from AUC (middle) to TUC1 (right), resulting in a shortening of the $c$-axis from 9.3 to 7.9 nm and a change in the $\beta$-angle from 94.5° to 90.6°.

population of DED peaks (particularly positive ones) is much greater for apo2 (Fig. 4b, left). The differences are primarily clustered at the bottom of the three-way junction, specifically latch residues 46–48, hinge residues 21–23, and junction residues 73–74 that connect P3 and P1.

The conversion of apo2 to IB·Ade, in which U48 is displaced by Ade[14], is strongly supported by the large (4.5 sigma) negative DED peak between U48's uracil base and sugar moiety. The further conversion of IB·Ade to B·Ade-like is impeded by the lattice constraint between the latch region in P2 and the P3-kissing loop of an adjacent symmetry molecule, and serves as the primary rate-limiting factor in the phase transition from AUC to TUC1. A21 also restricts the exit of U48 (latch) from the binding pocket. A21 is part of the flexible hinge, and ultimately forms one of three base triples that stabilize P1 in the B·Ade-like configuration. Together,

the constraints on IB·Ade in the AUC lattice limit molecular motion for quite some time, as minimal differences are observed between 10 and 25 s (Fig. 4b, right). The DED at G44, however, may reflect disruption of the dominant P2–P2 crystal packing interface in AUC (Fig. 4b). With highly redundant sets of diffraction data, at multiple time points during the AUC–TUC1 transition, we expect that the large conformational changes can be time-resolved crystallographically[27] and that transitional structures could be further constrained by an analysis of the diffusive scattering[23].

Using a combination of methods—PVM, AFM, and XFEL—the results of this study demonstrate, in real time, the synchrony of molecular changes that occur in the ligand-induced phase transition of riboA crystals (Supplementary Table 6). We also performed a similar PVM experiment using a caged adenine compound

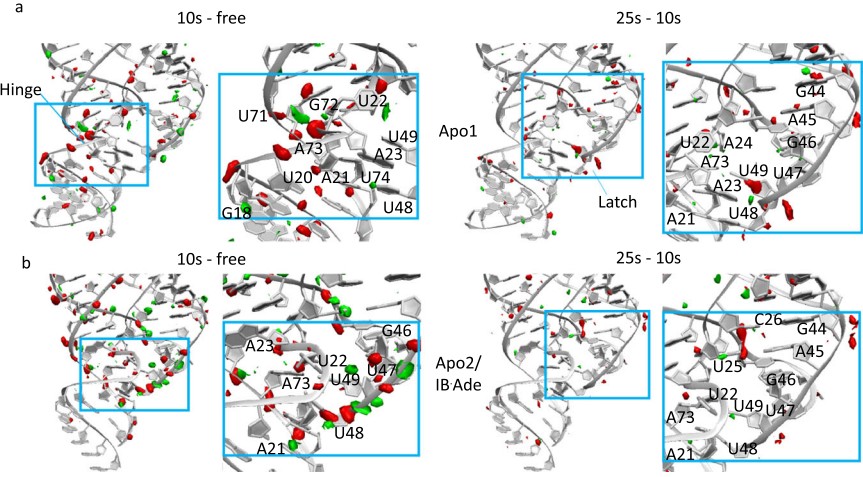

**Fig. 4 Difference electron density reflecting early stages of AUC–TUC1 transition.** $F_o - F_o$ difference electron density maps, displayed for apo1 (**a**) and apo2 (**b**), generated using the ligand-free model (PDB: 5E54) and the observed structure factor differences between data sets. Left: AUC-10 s—AUC free; right: AUC-25 s–AUC-10 s. Positive and negative density are shown in green and red, respectively, contoured at 3 sigma.

(Supplementary Fig. 6 and see also "Methods"), whereby the reaction of ligand binding was triggered by photoactivation at 365 nm (Supplementary Movie 3). Such experiments present an alternative approach to achieve potentially faster and more uniform reaction induction, particularly in cases where diffusive mixing is problematic. The synchronized SS transition in the riboA crystals is not unique, as revealed by PVM experiments using crystals of the tetracycline riboswitch aptamer (Supplementary Fig. 7). Therefore, the types of approaches presented here may prove highly useful for time-resolved studies of other biomolecular systems involving large conformational changes or crystal phase transitions.

## Methods

**RNA sample preparations**. Detailed experimental procedures for the in vitro transcription, purification, characterization, and crystallization of the adenine riboswitch aptamer RNA have been reported previously[14,28]. Crystallization buffer was prepared as 40 mM sodium cacodylate, 80 mM KCl, 100 mM MgCl₂, 12 mM spermine tetrahydrochloride, and 35–65% (v/v) 2-methyl-2,4-pentanediol (MPD). The stabilization buffer (SB) was prepared by mixing an equal volume of crystallization buffer and RNA buffer (RB: 10 mM HEPES pH 7.5, 10 mM KCl, 0.5 mM EDTA). All solutions were mixed to homogeneity prior to use, and stored at room temperature.

**Synthesis and characterization of the caged adenine N-(2-nitrobenzyl)-9H-purine-6-amine**. Photo-caged adenine (pcADE) was prepared as follows: 6-chloropurine (4.10 g, 26.5 mmol) and 2-nitrobenzylamine hydrochloride (5.00 g, 26.5 mmol) were suspended in 100 ml of isopropanol. Triethylamine (5.37 g, 53.0 mmol) was added and the reaction was warmed in a 90 °C oil bath. After 2 h, the temperature was reduced to 80 °C and the reaction was stirred for an additional 16 h. A dense precipitate formed during that time. The suspension was chilled on ice for 1 h and filtered. The filtrate was washed with several portions of cold isopropanol and dried under high vacuum to give 3.28 g (12.1 mmol, 45.7%) of N-(2-nitrobenzyl)-9H-purine-6-amine. Proton nuclear magnetic resonance (¹H NMR) (DMSO-$d_6$) δ = 12.99 (s, 1H, H), 8.28 (br s, 1H, H), 8.16–8.14 (m, 2H, H), 8.04 (d, J = 8.1 Hz, 1H, H), 7.66 (t, J = 7.6 Hz, 1H, H), 7.57 (d, J = 6.8 Hz, 1H, H), 7.50 (t, J = 7.7 Hz, 1H, H), 4.98 (s, 2H, H). ¹³C NMR (DMSO-$d_6$) δ = 154.1 (br), 152.3, 149.8 (br), 148.1, 139.3, 135.2 (br), 133.7, 129.9, 128.9, 124.5, 118.9 (br), 40.3 (br). MS (atmospheric pressure chemical ionization (APCI)) m/z = 271.1 [M + H]⁺.

NMR spectra were recorded on a Bruker spectrometer at 500 MHz for ¹H and 126 MHz for ¹³C. The following abbreviations were utilized to describe peak patterns: br = broad, s = singlet, d = doublet, t = triplet, and m = multiplet. All chemicals were obtained from Sigma-Aldrich and used without further purification. The identity and homogeneity of pcADE were assessed with a single quadrupole LC/MSD (Agilent) equipped with an in-line diode-array ultraviolet (UV) detector. Samples were injected onto a Zorbax rapid-resolution reversed-phase C18 column (bore size: 100 × 2.1 mm²; particle size: 3.5 μm), coupled with a C18 guard column (12.5 × 2.1 mm²). The contents were eluted at a flow rate of 300 μL/min with a 5–90% gradient of methanol/water containing 0.1% acetic acid. Samples were analyzed using APCI. Full scan mass spectra, as well as the total-ion chromatogram and the UV chromatogram at 270 nm, were used to assess compound purity.

**Polarized video microscopy**. In PVM, the polarization angle of linearly polarized light changes as it passes through a birefringent sample such as a crystal. An analyzer polarizer placed in front of a camera selects a single polarization angle for detection. Spatial and temporal differences in birefringence are detected as changes in transmitted light intensity at the camera, thus providing direct observation of the optical effects induced by the lattice transitions (Fig. 1). In order to perform a PVM experiment, we had to overcome the problem of crystal motion after mixing with the ligand, caused by turbulence as well as crystal shape changes. We tested several compounds to optimize crystal adhesion to glass surfaces for imaging. Initially, crystals grown by batch crystallization methods were harvested, adsorbed on glass-bottom dishes treated with poly-D-lysine, Cell-Tak, or just plasma cleaned, and then used for imaging. Ultimately, mica-grown crystals adhered to poly-D-lysine-coated glass were used for the final PVM experiments (discussed below in detail).

Supplementary Figure 1a illustrates the setup of the PVM. We modified a Nikon Eclipse TE200 from a surplus. The setup consists of a frame and base of a Nikon Eclipse TE200 microscope equipped with a halogen lamp light source for trans-illumination. Nikon microscope objectives ×100/1.45 Plan Apo Lambda and ×20/0.75 Plan Apo were used with a NA 0.72 condenser. Videos were recorded on a C-mounted OMAX color CMOS camera (Model A35180U3, AmScope, Irvine, CA) with 18 MP pixel resolution and the provided Touplite software that interfaces with the camera. A fixed polarizer in the "emitter" position functioned as an analyzer in the polarization setup. To implement computer-controlled rotatable polarizer, motorized rotational stage (Thorlabs, product # KPRM1E) was positioned above the condenser. Prior to the time-lapse imaging, fields of riboswitch crystals were identified by eye, and the linear polarizer was rotated to maximize the transmitted intensity of a majority of crystals in the field. Imaging was performed with the ×100 objective.

The adenine ligand was introduced to the crystal using two complementary methods. In the diffusion method, crystals grown on mica were extracted with 5 μL SB and deposited onto a poly-D-lysine-coated glass-bottom dish (MatTek Corporation, P35GC-1.5-14-C) (Supplementary Fig. 1b), and then covered with 1.5 mL SB. A measure of 1.5 mL of adenine (2 mM) in SB was added to a final concentration of 1 mM and video of the crystal was recorded (Supplementary Movie 1) with a 200 ms exposure time. A 30 × 30 pixel area in the yellow square box (Fig. 1a) before (left) and after (right) the phase transition is shown in Supplementary Fig. 1c. The pixel resolution of these oversampled images is 24.75 nm, but the theoretical diffraction limit for the system is 210 nm, or an area equivalent to ~9 × 9 pixels. The images are color-enhanced for clear visualization of individual pixels and the nonuniform crystal surface.

For the photoactivated experiments, a Thorlabs high-power UV LED at 365 nm (ThorLabs, Model SOLIS-365C) light source was installed at the backport of the microscope. The LED was powered by a Thorlabs driver (ThorLabs, Model DC2200) for epi-illumination with a UV beam. The filter cube contains a DAPI dichroic mirror, which directed UV-led light from the epi-port to the sample for the photoactivated experiments. Mica-grown crystals were extracted with 5 μL SB, and then deposited onto a glass-bottom dish. Two hundred microliters of 200 μL of freshly prepared pcADE (10 mM, dissolved in SB) was added to the crystals, and the glass-bottom portion of the dish was sealed with a greased coverslip (Supplementary Fig. 1b). The crystal was exposed with a 2-s LED pulse (4.5 A) of UV (365 nm) light, and the phase transition was recorded with an exposure time of 200 ms (Supplementary Movie 3).

**Analysis of polarization microscopy time-lapsed data**. Time-lapse polarization microscopy data were processed and analyzed using custom Matlab (ver. 2019b,

Mathworks) routines to identify when crystal lattice transitions occurred after ligand addition (or uncaging) and the speed of the transitions. The time-lapsed data consisted of a series of RGB images collected over the courses of the transitions and saved as an AVI file. Raw RGB AVI files from the polarization microscopy system were converted to greyscale uncompressed AVI files. The pixel resolution of the images is 24.75 nm, but the theoretical diffraction limit for the system $\Delta x = \lambda/2NA$, where $\lambda$ is the wavelength of light (taken as 500 nm) and NA is the numerical aperture of the objective (1.45 for the system used here), is 172 nm. $I(x,y,t)$ traces were smoothed in time using a sliding window of 15.2 s (39 time steps), prior to the derivative calculation. For the analysis presented here, a ROI of 180 pixels square (4.455 $\mu m^2$) was selected. We examined the effect of spatially resampling the ROI at each time point using sampling areas of 1, 3, 6, 9, 15, 30, 45, and 90 pixels square resulting in ROIs with 180, 60, 30, 20, 12, 6, 4, and 2 "superpixels" per side. A sampling area of 9 pixels corresponds to 225 nm, slightly greater than the theoretical diffraction limit for the polarization video microscope.

The recorded intensity for a given $(x,y)$ pixel at time $t$, $I(x,y,t)$, depends on the birefringence at that pixel relative to $t = 0$. For the crystals, $I(x,y,0)$ is a maximum, and $I(x,y,t)$ generally decreases as the crystal lattice undergoes conformational rearrangements and phase transitions that change the birefringence at each pixel. To identify phase transitions (sharp decreases in the $\partial I(x,y,t)/\partial t$), we calculated the first derivative of the intensity vs. time, $\partial I(x,y,t)/\partial t$. $\partial I(x,y,t)/\partial t$ has local minima at the transitions, and we multiplied $\partial I(x,y,t)/\partial t$ by $-1$ to make the data compatible with peak-finding algorithms that search for maxima in Matlab library. We identified and fit peaks in $-\partial I(x, y, t)/\partial t$ using the findpeaksxw.m routine (Pragmatic Introduction to Signal Processing Applications in scientific measurement, Tom O'Haver, March 2020 edition) with the following parameters: slope threshold, 0.00001; amplitude threshold, 0.6; smooth width, 50; fit width, 39; smooth type, 3; peak group, 2. Peak positions in time correspond to the transition times, and the peak half-widths at half-maxes correspond to the transition durations. For each sampling area size, transition times and durations were calculated for all superpixels in the ROI, and a $k$-means algorithm was used to classify the peaks into the clusters representing the three transitions in the $I(x,y,t)$ curves. Transition times and durations were averaged and plotted against sampling area to demonstrate scale invariance of the transition characteristics. The time where $\partial I_i[(xy)t)]/\partial t = 0$, the peak of the derivative for each pixel, is taken as the transition time.

Each peak in $-\partial I_i[(xy)t)]/\partial t$ was fit to a Gaussian. $K$-means clustering was used to identify three clusters (red, black, and blue) corresponding to the three main transitions in the crystal, namely, T1, T2 ($a$ and $b$), and T3. The time-lapsed video of the first transition was based on the ROI resampled with the 9 pixels square sampling area. For the video, the intensities for each pixel were normalized such that the minimum value was set to zero, and the maximum was normalized to the value at $I(x,y,t = 0)$. A false-color colormap was used to highlight the transitions. The $-(\partial I(x,y,t)/\partial t)$ frames are a binary representation of the transition with pixels with $-(\partial I(x,y,t)/\partial t)>$ amplitude threshold colored in red, and all others in blue. Sampling areas of 1, 3, 6, 9, 15, 30, 45, and 90 pixels square were used to spatially average "blocks" of the ROI. For each sampling area size, $k$-means clusters of peak half-width durations were calculated and averaged. Red, black, and blue circles correspond to first, second, and third transitions. Data are plotted mean ± standard deviation.

**Crystal growth on mica.** The AFM experiments required the riboA crystals to remain stationary throughout the imaging of the crystal before, during, and after the infusion of ligand and the lattice transition. DNA and RNA are routinely imaged with an AFM after absorption to mica treated with amino silanes. Mica is a standard, atomically flat AFM substrate with a silanol surface chemistry similar to glass. The electrostatic attraction between the negatively charged phosphate backbones of the nucleic acids and the positively charged amino groups of the substrate pins the molecules to the substrate. We reasoned that the riboswitch crystals would have negative surface charge density and that the crystals would likewise adhere electrostatically to a positively charged surface.

For the AFM experiments, we used mica treated with 3-aminopropylsilatrane (APS) as a positively charged substrate. APS-mica is a common solid surface for nucleic acid imaging by AFM because it has a lower surface roughness than mica treated with silanes such as (3-aminopropyl) triethoxysilane, which crosslink in addition to covalently binding to mica[29]. Due to the insufficient adsorption (possibly due to the crystal faces not being atomically flat and parallel to the mica surface across the entire crystal) and the physical changes in the crystal during the phase transition often desorbed the crystal from the mica surface. Moreover, the infusion of ligand solution through perfusion tubes creates a strong shear force on the crystals on the mica surface, which desorbs crystals. To overcome these two main issues, riboA crystals were grown directly on V1-grade mica surface by the hanging-drop vapor diffusion method. Mica itself is perfectly flat, pristine crystal free of inclusions or other defects, which may facilitate the riboA crystal growth evenly on mica surfaces by providing highly effective nucleation sites. We use seed crystals to initiate a large flat crystal growth on mica. To effectively absorb the crystal seeds on the surface, freshly peeled mica surfaces were first treated with 50 $\mu L$ of 100 mM spermidine for 15 min at room temperature. After surface treatment, the excess of spermidine on the mica surface was washed out with plenty of distilled water and air-dried. The spermidine-treated mica samples were then fixed on *EasyXtal* crystallization supports (Qiagen, Germantown, MD) using

paraffin oil. A measure of 0.5 $\mu L$ of 1:1000 dilution of riboA crystal seed stock was carefully placed on treated mica and incubated for 30 s. Without drying the seed solution, 5 $\mu L$ of an equal volume of 7.5 g/L gel-purified riboA in RNA buffer and 35% MPD crystallization buffer mixture was gently placed on the seed solution on mica samples. The support containing the crystallization sample solution was inverted and screwed onto the *EasyXtal* crystallization plate, followed by incubation at 22 °C for 12 h.

We grew crystals on the mica surface with two different concentrations of seed solutions to obtain *ab*- or *ac*-face crystals. For growing the rectangular-shaped *ac*-face crystals, a 1:1000 dilution of riboA crystal seeds was used. For square-shaped crystals with *ab* face, a 1:100 dilution of seeds was used (Supplementary Fig. 3).

**AFM imaging.** The crystals were imaged in SB containing 35% MPD with a Cypher VRS AFM outfitted with a temperature-controlled stage and perfusion cantilever holder (Asylum Research, Oxford Instruments, Santa Barbara, CA, USA) in intermittent contact mode using BL-AC40TS-C2 "BioLever mini" cantilevers (nominal tip radius of 8 nm, spring constant of 0.09 N/m and resonance frequency in SB of ~12.5 kHz) at 15 °C. These levers have a high sensitivity (100 nm/V) and low applied forces during imaging, which is essential for imaging soft biological crystals. A V1-grade mica sample with crystals was first fixed on an AFM sample holder disc (Ted Pella, Redding, CA, USA) using epoxy resin. Twenty microliters of SB was placed on the mica-crystal sample to avoid drying during fixation and covered tightly. The cantilever holder was connected with perfusion tubing to inlet and outlet liquid reservoirs (1 mL syringes) of SB to maintain the fluid volume (100 $\mu L$) and for buffer exchanges during measurement. After placing the sample on the AFM stage, the thermal peak of the cantilever in SB was generated and used to determine the resonance frequency (~12 kHz). The AFM imaging temperature was set at 15 °C, where the crystals were stable to sustain high-frequency scanning by the AFM tips. All images were recorded at a scan rate of 12 Hz, 256 pixels in $x$ and $y$, and 100-nm scan size. First-order flattening was applied line-by-line during scanning to remove sample tilt artifacts. At 100-nm scan size, scan rates >12 Hz with 256 points and lines are not optimal for continuous recording because of tip-induced damage to the crystal surface. Scan sizes of >100 nm can also be used with higher scan rates. But for the high-quality crystal surface topography, a frame size <100 nm is recommended because the image quality has a direct impact on unit-cell parameters derivation.

Before the addition of ligand, the apo lattice was imaged in SB continuously for at least 25–50 frames to determine the quality of the cantilever and crystal lattice. For continuous imaging, a crystal with a sizeable flat surface in the central region was chosen. A ligand-containing SB in a syringe containing 150 $\mu M$ (for *ac* face) or 50 $\mu M$ (for *ab* face) adenine ligand was slowly infused into the chamber while the AFM images recorded continuously. Immediately after the infusion of ligand, thermal and mechanic disturbances, results of mixing and external force from infusion, rendered image recording impossible for the first few minutes. This is one of the other reasons that the low temperature (15 °C) and ligand concentrations (150 or 50 $\mu M$) were chosen to slow down the transitions. For all experiments, the AFM images were recorded until the crystal lattice transition reached the final bound state, with gaps where the cantilever contact with the crystal lattice was lost, especially at the time when the crystal underwent significant physical changes. Nevertheless, the scanner position for imaging was unchanged throughout the experiment. Whenever the cantilever lost contact with the crystal surface, the imaging was stopped, and the tip approached the crystals again in its original assumed position to continue recording the transition. The most important experimental part is to use already standardized imaging parameters whenever the contact is lost. In addition, the orientation angle of crystal lattice from apo to the bound state was identical throughout the experiment, which is evident in the coordinated phase transition. Sometimes, due to the viscous nature of SB, the cantilever loosens from its original position, which may give rise to slightly tilted image acquisition. The drastic physical changes during phase transition could also result in images slightly offset from the original position. However, they do not impact the final results presented in the manuscript. In addition, to minimize damages to crystals or the cantilever tips, especially during the phase transition stages where the crystal undergoes drastic z-axis changes, we used the intermittent contact mode for imaging. In general, careful consideration was also given to the balance among various factors, such as recording of high-quality topographic images while minimizing damages to the lattice surface or cantilever tips throughout the experiments.

**AFM of reverse transition.** In addition, we conducted experiments with 50 $\mu M$ ligand using *ac*-face crystals, and recorded the phase transitions in both the forward and reverse directions. Once the crystal lattice transition reached BUC, the reversal of the phase transition was achieved by removing the ligand from the crystals through extensive buffer exchanges in situ. This was achieved by replacing the buffer with a ligand-free buffer ten times (1 mL each) and left overnight in the AFM chamber at 15 °C along with the cantilever on the same spot where the bound state was imaged. After overnight night soaking, the ligand-free buffer was infused and exchanged with fresh buffer (1 mL each) every 15 min for at least another 5 h. At the end of extensive buffer exchanges, the reversed apo lattice was recorded. To minimize evaporation of the buffer during overnight soaking, the temperature was kept at 15 °C.

**AFM image processing**. The AFM images were processed using Scanning Probe Image Processor software (Image Metrology, Lyngby, Denmark). The digital resolution of images was set to 2024 pixels. The raw images were processed using either the fast Fourier transform (FFT) or the correlation averaging utilities to remove random noise. When FFT was used, the final images were calculated using the inverse FFT utility. The autocorrelation images were calculated either from the raw images directly or autocorrelated averaged images. The unit-cell dimensions were determined manually by choosing the most prominent peaks from the autocorrelation images in the contour mode to avoid miscalculation from auto-mated analysis. For the images containing the mixed-cell lattice, the unit-cell parameters were measured directly from the images. All AFM images were processed and cropped into $50 \times 50$ nm$^2$ from their original size ($100 \times 100$ nm$^2$).

**RiboA batch crystallization**. RiboA microcrystal preparation for XFEL experiments was similar to that described in ref. [14]. Briefly, purified riboA (5 g/L in RB) was mixed to homogeneity with an equal volume of CB. The mixture was then inoculated (3% v/v) with a concentrated crystal seed stock, and the crystals were grown for 2 h at room temperature (~22 °C) with constant gentle mixing. The plate-like crystals (<5 μm) were centrifuged at $1000 \times g$. The supernatant was removed, and the crystals were washed with SB. The crystals were centrifuged and washed again, and finally concentrated to a maximum density of $10^9$–$10^{10}$ crystals/mL. The fine granular precipitate, which forms during crystallization, could not be completely removed, even upon filtering (10 μm), making it difficult to maximize the density of the crystal slurry without impeding/clogging sample injection. This is most likely the cause for low overall hit rates (~1%).

**Diffusive mixing experiments with liquid injection**. Technical details of the mixing experiments have been described previously[14]. Additional data were collected at the Coherent X-ray Imaging endstation[30] at the Linac Coherent Light Source (LCLS) during beam times LP58 and LM95. The riboA crystal slurry was filtered through a 20-μm wire-mesh filter immediately prior to loading into a stainless-steel reservoir for sample injection, which was then placed in an anti-settling device. The sample reservoir, as well as large (10 mL) reservoirs containing DEPC (diethyl pyrocarbonate)-treated water, SB, or SB with 20 mM adenine ligand, was connected via switchyards to one of two high-performance liquid chromatography (HPLC) pumps. Reservoirs containing sample, SB, or water were connected to HPLC1/switchyard1. Reservoirs containing ligand, SB, or water were connected to HPLC2/switchyard2. The injectors were inserted into the vacuum chamber using the "nozzle rod" system[31]. Diffusive mixing with liquid injection was achieved using high-resolution 3D-printed microfluidic mixers and nozzles integrated upstream of the X-ray interaction region. Our miniaturized adaptation of the static Kenics mixer Design 10, with 200 μm inner diameter and five helical elements for laminar stirring, provided complete mixing in <100 ms for 1:1 ratio of sample and ligand flowrates[24]. The mixed sample was fed via a delay capillary to the double-flow focusing nozzle[25] Design 8[24] with an additional outer-sheath flow of ethanol that improves jet stability and prevents icing. The mixing delay, which is the time between a crystal's exposure to ligand and its exposure to X-rays, was fully controllable through accurate placement (distance) of the mixers, and the combined flowrates (velocity), which were monitored by flowmeters. The time of diffusion, which likely occurs on the millisecond timescale[32], is considered negligible.

**XFEL data processing**. All XFEL data were acquired at the LCLS with an X-ray photon energy of ~9.5 keV, pulse energy of ~3.2 mJ, pulse length of ~30 fs, and repetition rate of 120 Hz. Data were recorded on a Cornell-SLAC Pixel Array Detector with live diffraction feedback using OnDA[33]. *Cheetah* and *CrystFEL* ($v0.9.0 + b5de2753$) were used for hit finding and indexing of diffraction images, respectively[34–38]. Detector geometry was optimized using *CrystFEL geoptimiser*[39]. The *peakfinder8* algorithm in *CrystFEL indexamajig* was used to identify Bragg reflections, with a pixel value threshold of 150, a minimum of 2 pixels per peak and minimum SNR = 4. The diffraction patterns were indexed and integrated using *CrystFEL indexamajig*, using *XGANDALF*[26], *MOSFLM*, *ASDF*, and *DirAx* indexing algorithms with "low-tolerance" (LT) parameters: peak-finding SNR = 4 and unit-cell tolerances of 7% for cell axes and 3° for cell angles[35,36]. Finally, the data sets were combined into a single stream file, and the merged and scaled with two iterations in *CrystFEL partialator*. The partialities for scaling were fixed to a value of 1.

Plotting indexing rates for AUC and TUC1 as a function of delay time reveals complementary sigmoidal curves: the decrease in AUC-indexed crystals is accompanied by an increase in TUC1-indexed crystals (Fig. 3a and Supplementary Fig. 5). The curves in Fig. 3a resemble the birefringence intensity curves from the PVM experiments (Fig. 1b). By 75 s, the vast majority of crystals index as TUC1, whose indexing rate plateaus at ~50%. From this analysis, it is clear that the decline in total "indexability" of crystals is correlated with the lattice transition driven by riboA conformational changes upon ligand binding. Importantly, however, the hit-rate remains fairly constant (~1%), indicating that Bragg diffraction is maintained throughout the transition, and that the crystals never enter a liquid or disordered phase. To further investigate the indexability of the data from 25 to 175 s post mixing, diffraction patterns that could not be indexed as TUC1 with LT parameters were "reindexed" with XGANDALF using "high-tolerance" (HT) parameters:

peak-finding SNR = 3.5 and unit-cell tolerances of 20% for cell axes and 10° for cell angles (Supplementary Fig. 5 and Supplementary Table 3).

**3D merging**. Diffraction patterns for individual data sets were oriented and assembled in 3D reciprocal space according to unit-cell parameters using the index and orientation information provided by *CrystFEL* indexing algorithms[26,36]. Slices of the 3D reciprocal lattice were processed in *FIJI*[40] and are displayed in Fig. 3. Changes in the AUC-indexed diffraction patterns, particularly at low resolution, were observable as early as 10 s, and became more prominent at 25 s. 3D-merged patterns at 75 s exhibited similar trends as those at 25 s, but were difficult to characterize with so few patterns (<2000) that could be indexed as AUC at 75 s. In $a*b*$, a set of intense elongated peaks emerges, corresponding to the TUC1 lattice ($a = 50.3$ Å and $b = 25.2$ Å), indicating that a significant portion of single crystals exhibits both AUC and TUC1 diffraction. In the $a*c*$ plane, the marginal increase in the length of the $a$-axis was barely detectable, whereas the dramatic decrease in the $c$-axis (from 93.4 to 78.7 Å) and change in the beta-angle (from 94.5 to 90.6) was observed as azimuthal streaks about $b*$.

**DED map calculations**. Ligand-free and 10 s mixed data from our previously reported work[14] were reindexed using the same version of *CrystFEL* and indexing parameters as used for the data in the current study, yielding 29,052 and 23,786 indexed patterns, respectively (Supplementary Table 5). Those 10 s mixed data were then combined with the 10 s mixed data (14,321 indexed patterns) from the current experiment. The three data sets—AUC free (29,052 patterns), "combined" AUC-10 s (38,107 patterns), and AUC-25 s (10,357 patterns)—were merged and scaled in *CrystFEL partialator* using the *custom-split* option to ensure all data were put on the same scale before exporting the final reflection files for individual data sets. $F_o$–$F_o$ maps (Fig. 4) were generated with the *PHENIX* software suite[41] using the ligand-free model for phase calculations. Since the phosphate groups are more electron-rich, the majority of DED peaks reside along the RNA backbone. Overall, the net DED appears to be negative (more red than green), indicating that, in general, the transitioning molecules become less ordered, that is, the positive DED is dispersed. However, DED is more balanced for apo2, which exhibits more prominent positive peaks (Fig. 4b).

Unlike the binding pocket, there is minimal restriction on P1 mobility, which has been observed previously to have incomplete electron density and significantly elevated B-factors even in a ligand-free environment, and is particularly worse for the apo2 conformer[14]. The paucity of DED peaks in the P1 of apo2 (Fig. 4b) suggests that the regulatory helix is very dynamic at the onset of conformational switching. The mobility of residues A21, U22, and A23 in the hinge region, directly adjacent to the top of the P1 stem, most likely enable the release of ligand-induced energetic strain and partial unwinding of the helix. The combined motion in the three-way junction, however, propagates throughout the P3 helix and the 3′ strand of P1, observed as negative DED peaks (Fig. 4b). We suspect, therefore, that the displacement of U48, which affects the bottom of P3 via the three-way junction and the top of P3 via lattice interactions, may temporarily alter the stability of both P1 and P3, as the molecule transitions to B·Ade-like.

A similar analysis was done by comparing TUC1-indexed data from 25 to 75 s with that of 100 to 175 s. However, no discernable differences in electron density between the two data sets were observed. This further supports that the bulk conformational changes in reaching a B·Ade-like state may occur in the context of the AUC lattice. However, AUC would seem to require enough expansion to overcome lattice restraints inhibiting reconfiguration of the binding pocket—thereby enabling rotation and translation of the P1 helices, as both molecules coordinate to stabilize into the lower-energy TUC1 lattice. As a result, structure determination of intermediate conformers in the trajectories from apo1 to apo2, or IB·Ade to B·Ade-like, may be feasible by sorting of structure factor amplitudes and their associated phases through singular value decomposition[42].

**Structure determination and refinement**. Molecular replacement (MR) and model refinement were performed using the *PHENIX* software suite[41]. All structure analysis and modeling were done in *Coot*[43]. Images containing structural models were generated using The PyMOL Molecular Graphics System (Schrödinger, LLC). The crystal structures derived from TUC1-indexed (6VWT) or TUC2-indexed (6VWV) data were solved by MR using the riboA bound structure (RCSB: 4TZX) as a search model. Crystal data and refinement statistics are summarized in Supplementary Table 4.

## Data availability
Structure factors and model coordinates for the crystal structures of TUC1 and TUC2 are available from the PDB under accession codes 6VWT and 6VWV, respectively. Data regarding AFM and PVM experiments are available within the manuscript. Source data are provided with this paper.

## Code availability
The code for the analyses of PVM data is available for download from the GitHub site with link: https://github.com/Will-Heinz/OMAL-PVM-Analyzer.

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

## Acknowledgements
This work was supported in part by the Intramural Research Program of the National Cancer Institute, National Institutes of Health (Y.-X.W.), the NSF-STC "BioXFEL" (NSF-1231306) (J.C.H.S.), the Maxwell computational resources operated at Deutsches Elektronen-Synchrotron (DESY) (H.N.C.), Hamburg, Germany (H.N.C.), and NSF Award (NSF-1565180) (N.A.Z.). Use of the Linac Coherent Light Source (LCLS), SLAC National Accelerator Laboratory, is supported by the US Department of Energy, Office of Science, Office of Basic Energy Sciences under Contract No. DE-AC02-76SF00515 (S.B.). Parts of the sample injector used at LCLS for this research were funded by the National Institutes of Health, P41GM103393, formerly P41RR001209.

## Author contributions
Y.-X.W. conceptualized and designed the study; P.Y. and J.R.S. provided the RNA samples; P.Y., and J.D. characterized RNA samples; S.R. and J.R.S. crystallized the RNA; J.R.S., C.E.C., Y.R.B., O.Y., M.O.W., J.K., D.O., T.A.W., A.B., V.Mariani, C.L., W.B., M.S.H., S.B., S.P., M.S., T.D.G., N.A.Z., and Y.-X.W. performed TRX experiments and analysis of diffraction data; S.R., J.R.S., W.F.H., V.Magidson, S.L., and Y.-X.W. performed PVM experiments. W.F.H. wrote the MATLAB codes for the PVM analysis. Y.-X.W. W.F.H., S.R., J.R.S., and Y.-X.W. analyzed PVM data. S.R., W.F.H., and J.D. performed AFM experiments. Y.-T.L. provided tetracycline riboswitch data. G.P. synthesized caged adenine. Y.-X.W. and X.Z. performed kinetics analyses. J.C.H.S. and H.N.C. provided XFEL expertise and support. S.R., J.R.S., and Y.-X.W. drafted the manuscript and all authors contributed to the revision.

## Funding

## Competing interests
