## [Peer Review File · Nature Communications]

REVIEWER COMMENTS

Reviewer #1 (Remarks to the Author):

This paper shows very interesting results of using riboA crystals to capture RNA changes in response to the reaction, using SFX, AFM and PVM. In particular, the macroscopic movement of the AFM and PVM is coupled with the microscopic movement of the SFX, which is very useful for observing the structural changes. The results are presented in detail, from the experimental methods to the detailed data, and I recognize that the results are very useful.

In Fig4, authors calculate the difference electron density map of 10s-mixed - ligand-free, and 25s-mixed - 10s-mixed. I think that SFX statistic data of 10s- and 25s-mixed are necessary. In Extended Data Table4, SFXdata of TUC1 and TUC2 are shown. I am confused about the naming of TUC1, TUC2, 10s-mixed, 25s-mixed, etc.

What percentage of Caged group of caged Adenine is disconnected by one installation of 365nm UV LED? In other words, what is the quantum yield? Of course, in this experiment, if the quantum yield is high, there is no problem, but if it is low, does it cause any problems?

In Extended Data Table1, which data are AFM Figs. 2b and 2c?

Extended Data Fig.6

c, The time course of ... -> b, The time...

Reviewer #2 (Remarks to the Author):

Ramakrishnan and coauthors report observation and analysis of Synchronous RNA conformational changes trigger ordered phase transitions in crystals. In particular adenine ligand was used to induce the RNA conformational change and crystal phase transition. The study's conclusion is supported by multiple complementary methods including AFM, polarized microscopy and XFEL diffraction whose results are generally internally consistent. This is an important step toward rigorous understanding how macromolecule dynamics interplay with crystal phase transition to allow future capture of molecular large conformational changes using cutting-edge XFEL method.

Several comments listed below for consideration:

(1) One key question regarding the RNA ligand system, is the observed time scale of phase transition and conformational changes consistent with literature or expected physiological function time scale? In other words, is the observed kinetics on the 10 to hundreds of seconds scale fits the biological function?

(2) Would there be a physics boundary expected on time and distance scale where molecular motion can interplay with crystal phase transition? That will help design future XFEL experiments

(3) For mentioning the crystal become live and comparison to living organism phenomenon, it's a bit of stretch since this particular crystal example is artificial crystallization and the overall condition may be very well away from physiological. However, perhaps the intent was to emphasize the nanotechnology aspect, which shall be then be rephrased wherever applies.

Reviewer #3 (Remarks to the Author):

The paper by Ramakrishnan et al. reports large structural changes induced by ligand-binding, which they call solid-to-solid phase transitions, in the crystalline state of an adenine riboswitch aptamer RNA (riboA), studied by polarized video microscopy, atomic force microscopy and XFEL.

They showed that the structural changes induced lattice order changes in the crystalline state of the enzyme without destroying the lattice order itself, allowing them to study the structural changes in the crystalline state by the X-ray diffraction method using XFEL. This is made possible by the synchronous molecular rearrangements in the crystals. While this is one example, there may be opposite cases where large structural changes may destroy the lattice order, making time-resolved X-ray crystallography on such enzymes not possible. On the other hand, synchronous molecular rearrangements are expected, provided that the mixing of ligand and enzyme is rapid enough, irrespective of in vivo or in isolated biological systems, which may maintain the lattice order while changing its cell dimensions. The present study is apparently an example of the latter case. However, it does not provide new advancements on the enzyme reaction itself, and the structural changes of the enzyme have been reported in a previous paper by the same authors (6).

Some specific points are listed below.

1. Page 6: Three main transitions (T1, T2, T3) were described, "with T1 centered at 227 s with more than 90% of all pixels completing transition within 5 seconds". What does this mean? The transition time described here does not seem to correspond to the transition time described later resolved by XFEL, where TUC1 was already observed at 10 s (Page 10). Please explain.

2. In the abstract and conclusion sections, the synchronous behavior was discussed in relation to living systems. However, the synchronous behavior in living systems that the authors cited is related to the timing of the system that is controlled by other factors, whereas the phenomenon described in the manuscript solely depends on the mixing speed of the ligand with the enzyme, without any control of outside factors. Thus, these are different phenomena and cannot be compared.

3. Methods section, page 21, second paragraph: The sentence "The crystal (ac-type) shown in" was incomplete.

4. The structures of TUC1 and TUC2 were reported in the paper. However, giving the large changes in the cell parameters, the structures of AUC and BUC should also be reported and compared.

RESPONSES TO REVIEWER COMMENTS FOR MANUSCRIPT NCOMMS-20-30564-T

Reviewer #1:

1. In Fig4, authors calculate the difference electron density map of 10s-mixed - ligand-free, and 25s-mixed - 10s-mixed. I think that SFX statistic data of 10s- and 25s-mixed are necessary. In Extended Data Table4, SFXdata of TUC1 and TUC2 are shown. I am confused about the naming of TUC1, TUC2, 10s-mixed, 25s-mixed, etc.

Regarding the reviewer's comment about Fig. 4, a new table, now "Extended Data Table 5," has been added that contains the data processing statistics for the data sets used in the difference electron density map calculations: "AUC-free," "combined AUC-10s," and "AUC-25s." We thank the reviewer for pointing out this missing information. As for the reviewer's second comment about naming, we understand how this can be confusing. We have changed the nomenclature in the text (p. 11) for 10s-mixed and 25s-mixed to read AUC 10s-mixed (AUC-10s) and AUC 25s-mixed (AUC-25s), respectively. This should clarify to the reader that the difference density map analysis takes place in the AUC setting, as the phase transition to TUC1 ensues. In regard to the identity of TUC1 and TUC2, TUC1 is the resulting lattice after the first stage of the phase transition (from AUC) that occurs upon the large conformational changes associated with ligand binding. TUC2 is the second intermediate lattice observed in a small population of SFX data. The detailed explanations of the meaning of TUC1 and TUC2 are given in pages 8, 10-11.

2. What percentage of Caged group of caged Adenine is disconnected by one installation of 365nm UV LED? In other words, what is the quantum yield? Of course, in this experiment, if the quantum yield is high, there is no problem, but if it is low, does it cause any problems?

In our hands, the ligand (0.5 -1 mM) uncaging after 1-2 sec of exposure at 4500 mW, has a quantum yield in the range of 88 – 94 %. Thus, the efficiency of uncaging poses no issue for our experiments.

3. In Extended Data Table1, which data are AFM Figs. 2b and 2c?

We thank the reviewer for pointing out this error. These labels in Extended Data Table 1 have been corrected to reference Extended Data Fig. 4b and 4c.

***4. Extended Data Fig.6
c, The time course of ... -> b, The time...***

We thank the reviewer for pointing out this typographical error. We have corrected it.

Reviewer #2:

1. One key question regarding the RNA ligand system, is the observed time scale of phase transition and conformational changes consistent with literature or expected physiological function time scale? In other words, is the observed kinetics on the 10 to hundreds of seconds scale fits the biological function?

We thank the reviewer for bringing up this important distinction. The short answer to the question, is “no.” Since the ligand-induced changes are large and restricted by lattice restraints, the timescale of transition is longer in crystallo than in solution. These experiments are not intended to characterize the behaviors of the RNA molecules on the solution time scale. One of goals of this study is to determine the precise time window where the conformational transition occurs in crystals. This time window is essential for the XFEL experiments designed to catch the molecular movie of the large conformational changes in the right time frame. Thus the information obtained from our experiments paves the way to record the molecular movie of the RNA conformational change in slow motion using time-resolved crystallography and an XFEL. It is noteworthy to point out that in this sense, the time-resolved crystallography of a system undergoing large conformational changes is distinct from that involving minute conformational changes, such as those triggered by light.

2. Would there be a physics boundary expected on time and distance scale where molecular motion can interplay with crystal phase transition? That will help design future XFEL experiments.

The main purpose of our study is to analyze the synchronous rearrangements of the RNA molecules, the interplay between the RNA conformational transitions and the SS phase transitions in crystallo. The synchronous transitions make it possible to record interpretable diffraction data. Fig.1 illustrates the time course of the phase transition with an emphasis on the T1 transition, which correlates with the spatial lattice transition from AUC to TUC1 (Fig. 2). The molecular conformational transitions and the concurrent lattice transition are shown further to be correlated via X-ray diffraction in the forms of both raw data and molecular structures (Fig. 3). The time-course shown in Fig. 1 is particularly important for two reasons: 1). The PVM experiments were performed under ligand concentrations similar to that for the XFEL experiments. Therefore, the time-course experiments, combined with our kinetics model (Stagno et al., 2016) can be used to simulate the phase transition at different ligand concentrations to guide XFEL experiments. 2). As the large RNA conformational changes occur during the T1 transition, these experiments are paramount to future mix-and-inject experiments by defining the exact time window of the T1 transition. This question is relevant to the previous question.

3. For mentioning the crystal become live and comparison to living organism phenomenon, it's a bit of stretch since this particular crystal example is artificial crystallization and the overall condition may be very well away from physiological. However, perhaps the intent was

to emphasize the nanotechnology aspect, which shall be then be rephrased wherever applies.

We thank the reviewer for this comment and we understand the reviewer's concern. We did not intend to equate phase transitions of artificial crystals to naturally occurring phenomena. Our goal was to illustrate the commonality of synchronized behaviors, whether in natural or artificial crystals of inorganic or biological molecules, synthetic materials, or living systems. Importantly, the mechanism of a solid-solid phase-transition at the molecular level, naturally occurring or otherwise, involves synchronous changes (sometimes even large ones) without entering the liquid phase. As the reviewer points out, this may have important implications for materials science as well. To avoid misinterpretation, however, we have revised the last paragraph (p. 13) and removed language that compares artificial crystals to living systems.

Reviewer #3:

1. Page 6: Three main transitions (T1, T2, T3) were described, "with T1 centered at 227 s with more than 90% of all pixels completing transition within 5 seconds". What does this mean? The transition time described here does not seem to correspond to the transition time described later resolved by XFEL, where TUC1 was already observed at 10 s (Page 10). Please explain.

We thank the reviewer for this comment and hope to eliminate confusion. We observed the first transition T1 to be centered around 227 s, with a transition width (duration) of ~5 s, during which 90% of the pixels in the selected area completed the transition (based on intensity). We have added "(transition width)" to the end of the statement on p.6 to clarify that 5 seconds refers to the duration of the T1 transition. The reviewer is correct that the transition times for T1 measured in the PVM, AFM, and XFEL experiments are not similar. This is due primarily to the large differences in the optimal ligand concentrations used in each method (PVM: 1mM, AFM: 0.05-0.15 mM, XFEL: 10mM), which has a large effect on the kinetics of the phase transition. This is addressed in the caption for Fig. 3. However, we added the ligand concentration used in PVM to the caption for Fig. 1. We should also point out that technical limitations of the PVM, AFM, and XFEL experiments dictate the concentration ranges that can be feasibly used for the measurements. The concentrations between PVM and XFEL experiments, at least, are similar (1 mM vs 10 mM), thus enabling us to simulate and test the kinetics of the phase transition at different ligand concentrations in the range of 1-10mM, which is crucial for guiding future mix-and-inject experiments using an XFEL.

2. In the abstract and conclusion sections, the synchronous behavior was discussed in relation to living systems. However, the synchronous behavior in living systems that the authors cited is related to the timing of the system that is controlled by other factors, whereas the phenomenon described in the manuscript solely depends on the mixing speed of the ligand with the enzyme, without any control of outside factors. Thus, these are different phenomena and cannot be compared.

We thank the reviewer for this comment and we understand the reviewer's concern. Our goal was to illustrate the commonality of (not directly compare) synchronized behaviors, whether in natural or artificial crystals of inorganic or biological molecules, synthetic materials, or living systems. Importantly, the mechanism of a solid-solid phase-transition at the molecular level, naturally occurring or otherwise, involves synchronous changes (sometimes even large ones) without entering the liquid phase. We did not intend to extend the conclusions of this study to other naturally occurring phenomena. To avoid misinterpretation, we have revised the last paragraph (p. 13) and removed language that compares artificial crystals to living systems. The last sentence of the abstract, however, was left alone, as this statement merely communicates that such synchronous behavior, which is observed in nature, is not limited to living systems.

3. Methods section, page 21, second paragraph: The sentence "The crystal (ac-type) shown in" was incomplete.

We thank the reviewer for pointing out this typographical error. The missing text has been added.

4. The structures of TUC1 and TUC2 were reported in the paper. However, giving the large changes in the cell parameters, the structures of AUC and BUC should also be reported and compared.

We thank the reviewer for this comment. Indeed, the changes in cell parameters among all four lattices are significant. These are reported in Extended Data Table 1. However, the crystal structures of AUC and BUC, and their corresponding crystal data and refinement statistics have been reported previously (Stagno *et al.*, 2017). Therefore, the crystal data and refinement statistics reported in this study includes only the novel structures, TUC1 and TUC2. We have, however added a table, now "Extended Data Table 5", which includes data reduction statistics for the data sets used in difference density map calculations.

References

Stagno, J. R., Liu, Y., Bhandari, Y. R., Conrad, C. E., Panja, S., Swain, M., Fan, L., Nelson, G., Li, C., Wendel, D. R., White, T. A., Coe, J. D., Wiedorn, M. O., Knoska, J., Oberthuer, D., Tuckey, R. A., Yu, P., Dyba, M., Tarasov, S. G., Weierstall, U., Grant, T. D., Schwieters, C. D., Zhang, J., Ferre-D'Amare, A. R., Fromme, P., Draper, D. E., Liang, M., Hunter, M. S., Boutet, S., Tan, K., Zuo, X., Ji, X., Barty, A., Zatsepin, N. A., Chapman, H. N., Spence, J. C., Woodson, S. A. & Wang, Y. X. (2017). *Nature* **541**, 242-246.

REVIEWERS' COMMENTS

Reviewer #1 (Remarks to the Author):

In this paper, the combination of PVM, AFM and XFEL has been used to successfully capture changes in the crystal of biomacromolecules and the movement of molecules in the crystal. The relationship between the microscopic movement of the molecules in the crystal and the macroscopic changes in the crystal itself is shown to be very clear. This is an excellent paper that suggests a new research method for the future.

Reviewer #2 (Remarks to the Author):

The authors have properly addressed my concern with explanation and additional technical information.

One minor suggestion:

If slow motion enabled by crystal lattice to capture large conformational change is desirable for this kind of experiment, perhaps this rationale shall be mentioned in writing and clarified to be distinct from physiological functional time scale (if known)

Reviewer #3 (Remarks to the Author):

The revised manuscript by Ramakrishnan et al. has addressed my concerns adequately. I have only a few grammatical questions that need the authors' attention.

1. Page 5, line 3 – line 2, page 6: "Detailed analyses at the pixel level of the digitized video (Fig. 1a, yellow square of ROI), which contains ~400 million RNA molecules, is about the typical size of crystals used in the XFEL experiments". What does this sentence mean?
2. Page 6, lines 4-7: "The time-traced transmitted light intensity vs. time of a single pixel exhibits multiphasic sigmoidal transitions with ~50% total loss in transmitted light intensity $I_i[(xy)_i,t]$ of pixel I ($i=1,2,...32400$) (Fig. 1b (left)).". When or at what stage does "~50% total loss in transmitted light intensity" occur?
3. Page 10, line 1: "...aligned TRX in three-dimensional space". Is TRX necessary here?

RESPONSES TO REVIEWER COMMENTS FOR MANUSCRIPT NCOMMS-20-30564-T

Reviewer #1 (Remarks to the Author):

In this paper, the combination of PVM, AFM and XFEL has been used to successfully capture changes in the crystal of biomacromolecules and the movement of molecules in the crystal. The relationship between the microscopic movement of the molecules in the crystal and the macroscopic changes in the crystal itself is shown to be very clear. This is an excellent paper that suggests a new research method for the future.

We thank Reviewer #1 for his or her comments.

Reviewer #2 (Remarks to the Author):

The authors have properly addressed my concern with explanation and additional technical information.

One minor suggestion:

If slow motion enabled by crystal lattice to capture large conformational change is desirable for this kind of experiment, perhaps this rationale shall be mentioned in writing and clarified to be distinct from physiological functional time scale (if known)

We thank Reviewer #2 for his or her comments and suggestion. We have added a sentence in the last paragraph of the introduction: “Due to lattice restraints, the conformational changes that occur in crystal, which drive the crystal phase transition, take place on a timescale much greater than that which occur in solution or under physiological conditions.”

Reviewer #3 (Remarks to the Author):

The revised manuscript by Ramakrishnan et al. has addressed my concerns adequately. I have only a few grammatical questions that need the authors' attention.

1. Page 5, line 3 – line 2, page 6: “Detailed analyses at the pixel level of the digitized video (Fig. 1a, yellow square of ROI), which contains ~400 million RNA molecules, is about the typical size of crystals used in the XFEL experiments”. What does this sentence mean?

We thank Reviewer #3 for his or her comment. We agree with the Reviewer that the sentence is confusing as written. We have revised the text to read as follows: “Detailed analyses were performed at the pixel level for a region of interest (ROI, Fig. 1a, yellow square) of the digitized video. The ROI, which contains ~400 million RNA molecules, is about the typical size of crystals used in the XFEL experiments⁶.”

2. Page 6, lines 4-7: “The time-traced transmitted light intensity vs. time of a single pixel exhibits multiphasic sigmoidal transitions with ~50% total loss in transmitted light intensity $I_i[(xy)_i,t]$ of pixel I

($i=1,2,..32400$) (Fig. 1b (left)).” . When or at what stage does “~50% total loss in transmitted light intensity” occur?

We thank Reviewer #3 for his or her comment. We have added a clause to the end of the sentence to provide clarification. It now reads as follows: “The time-traced transmitted light intensity vs. time of a single pixel exhibits multiphasic sigmoidal transitions with ~50% total loss in transmitted light intensity $I_i[(xy)_i, t]$ of pixel i ($i=1,2,..32400$) (Fig. **1b** (left)), over the course of the entire transition.”

3. Page 10, line 1: “...aligned TRX in three-dimensional space”. Is TRX necessary here?

We thank Reviewer #3 for pointing out this typographical error. We have removed “TRX.”